# BETTER GENERALIZATION WITH ON-THE-FLY DATASET DENOISING

## ABSTRACT

Memorization in over-parameterized neural networks can severely hurt generalization in the presence of mislabeled examples. However, mislabeled examples are to hard avoid in extremely large datasets. We address this problem using the implicit regularization effect of stochastic gradient descent with large learning rates. By the loss statistics, we are able to identify mislabeled examples with remarkable success. Then, we discard the mislabeled examples on the fly and continue training with the rest of the examples. This leads to ON-THE-FLY DATA DENOISING (ODD), a simple yet effective algorithm that is robust to mislabeled examples, while introducing almost zero computational overhead. Empirical results demonstrate the effectiveness of ODD on several datasets containing artificial and real-world mislabeled examples.

## 1 INTRODUCTION

Over-parametrized deep neural networks have remarkable generalization properties while achieving near-zero training error (Zhang et al., 2016). However, the ability to fit the entire training set is highly undesirable, as a small portion of mislabeled examples in the dataset could severely hurt generalization (Zhang et al., 2016; Arpit et al., 2017). Meanwhile, an exponential growth in training data size is required to linearly improve generalization in vision tasks (Sun et al., 2017); this progress could be hindered if there are mislabeled examples within the dataset.

Mislabeled examples are to be expected in large datasets that contain millions of examples. Web-based supervision produces noisy labels (Li et al., 2017a; Mahajan et al., 2018); whereas human labeled datasets sacrifice accuracy for scalability (Krishna et al., 2016). Therefore, algorithms that are robust to various levels of mislabeled examples are warranted in order to further improve generalization for very large labeled datasets.

In this paper, we propose *On-the-fly Data Denoising* (ODD), a simple and robust method for training with noisy examples based on the implicit regularization effect of stochastic gradient descent. First, we train residual networks with large learning rate schedules and use the resulting losses to separate clean examples from mislabeled ones. This is done by identifying examples whose losses exceed a certain threshold. Reasonable thresholds can be derived from the loss distribution for uniform label noise which does not depend on the amount of mislabeled examples in the dataset. Finally, we remove these examples from the dataset and continue training until convergence.

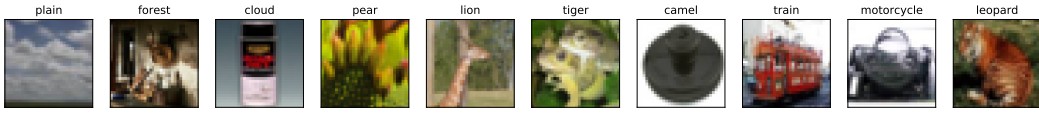

Figure 1: Mislabeled examples in the CIFAR-100 training set detected by ODD.

Empirically, ODD performs significantly better than previous methods in datasets containing artificial noise (Sections 4.1 and 4.2) or real-world mislabeled examples (Section 4.3), while achieving equal or better accuracy than the state-of-the-art on clean datasets (Sections 4.1 and 4.2). We further conduct ablation studies to demonstrate that ODD is robust w.r.t hyperparameters and artificial

noise levels (Section 4.4). Our method is also able to detect mislabeled examples in the CIFAR-100 dataset without any additional supervision (Figure 1).

## 2 DENOISING DATASETS ON-THE-FLY

The goal of supervised learning is to find a function $f \in \mathcal{F}$ that describes the probability of a random label vector $Y \in \mathcal{Y}$ given a random input vector $X \in \mathcal{X}$, which has underlying joint distribution $P(X, Y)$. Given a loss function $\ell(\mathbf{y}, \hat{\mathbf{y}})$, one could minimize the average of $\ell$ over $P$:

$$\mathcal{R}(f) = \int \ell(\mathbf{y}, f(\mathbf{x})) \, dP(\mathbf{x}, \mathbf{y})$$

The joint distribution $P(X, Y)$ is usually unknown, but we could gain access to its samples via a potentially noisy labeling process, such as crowdsourcing (Krishna et al., 2016) or web queries (Li et al., 2017a). We denote the training dataset with $N$ examples as $\mathcal{D} = (\mathbf{x}_i, \mathbf{y}_i)_{i \in [N]} = \mathcal{G} \cup \mathcal{B}$. $\mathcal{G}$ represents correctly labeled (clean) examples sampled from $P(X, Y)$. $\mathcal{B}$ represents mislabeled examples that are not sampled from $P(X, Y)$, but from another distribution $Q(X, Y)$; $\mathcal{G} \cap \mathcal{B} = \varnothing$. We aim to learn the function $f$ from $\mathcal{D}$ without knowledge about $\mathcal{B}, \mathcal{G}$ or their statistics (e.g. $|\mathcal{B}|$).

A typical approach is to assume that $\mathcal{B} = \varnothing$, i.e. all examples are i.i.d. from $P(X, Y)$, and minimizing the following objective:

$$\hat{\mathcal{R}}(f) = -\frac{1}{N} \sum_{i=1}^{N} \ell(\mathbf{y}, f(\mathbf{x}))$$

If $\mathcal{B} = \varnothing$ is indeed true, then $\hat{\mathcal{R}}(f) \to \mathcal{R}(f)$ as $N \to \infty$. However, this is not true if $\mathcal{B} \neq \varnothing$ since $\mathcal{D}$ is no longer an unbiased population of $P(X, Y)$. Moreover, when $\mathcal{F}$ is the space of large neural networks with parameters exceeding $N$, $f$ could fit the entire training dataset (Zhang et al., 2016), including the mislabeled examples. This results in undesired behavior of $f$ on inputs of the mislabeled set, let alone outside the training data.

To illustrate the harm of mislabeled examples to generalization, we consider training on CIFAR-10 where some examples are mislabeled uniformly at random. Compared with training on $\mathcal{D}$, training only on $\mathcal{G}$ could decrease validation error from 11.53 to 4.25 if there are 20% mislabeled examples, and from 15.57 to 5.06 if there are 40% mislabeled examples[1]. Therefore, if we are able to identify examples that belong to $\mathcal{G}$, we could vastly improve generalization on $P(X, Y)$.

### 2.1 SEPARATING MISLABELED EXAMPLES VIA IMPLICIT REGULARIZATION

Fortunately, in the case of classification with deep residual networks (He et al., 2015a), the implicit generalization of stochastic gradient descent (SGD) with large learning rates (e.g. 0.1) can separate examples from $\mathcal{G}$ and examples from $\mathcal{B}$ via the loss statistics. We demonstrate this in Figure 2, where we train deep residual networks on CIFAR-100 and ImageNet with different percentages of uniform label noise. In early stages of training, the loss distributions of clean examples and mislabeled ones have notable statistical distance. The network starts to fit mislabeled examples when learning rate starts to decrease, which is also crucial for achieving better generalization on clean datasets.

The working of the implicit regularization of gradient descent is by and large an open question that attracts much recent attentions (Neyshabur, 2017; Li et al., 2017b; Du et al., 2018). Empirically, it has been observed that large learning rates are beneficial for generalization (Kleinberg et al., 2018). Recent work has shown that the stationary distribution of SGD iterates corresponds to an Ornstein-Uhlenbeck process (Uhlenbeck & Ornstein, 1930) with noise proportional to the learning rate (Mandt et al., 2017). Training with large learning rates would then encourage solutions that are more robust to large random perturbations in the parameter space and less likely to overfit to mislabeled examples. Therefore, given these empirical and theoretical evidence on large learning rate helps generalization, we propose to classify correct and mislabeled examples through the loss statistics, and achieve better generalization by removing the examples that are potentially mislabeled and training on clean examples only.

---

[1]Validation error is 3.73 for 0% mislabeled case. Additional details for CIFAR-100 in Table 1.

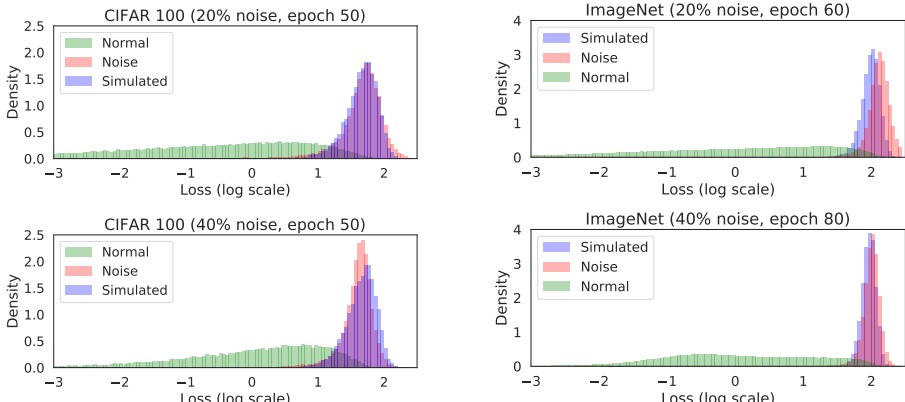

Figure 2: Histogram of the distributions of losses, where "normal", "noise", and "simulated" denote normal examples, examples corrupted with random labels and simulated losses respectively.

## 2.2 THRESHOLDS THAT CLASSIFY MISLABELED EXAMPLES

To improve generalization in practice, one critical problem is to select a reasonable threshold for classification. High thresholds could include too many examples from $\mathcal{B}$, whereas low thresholds could prune too many examples from $\mathcal{G}$; reasonable thresholds should also adapt to different unknown ratios of mislabeled examples. Let us first consider the case where $Q(Y|X)$ has the highest entropy, which is the uniform distribution over labels. From Figure 2, the loss distribution for $\mathcal{B}$ is relatively stable with different ratios of $|\mathcal{B}|/|\mathcal{D}|$; examples in $\mathcal{B}$ are making little progress when learning rate is large. We propose to characterize the (negative log-likelihood) loss distribution of uniform label noise $p_n(l)$ via the following generative procedure:

$$l = -\tilde{\mathbf{y}}_k + \log\left(\sum_{i\in[N]}\exp(\tilde{\mathbf{y}}_i)\right) \equiv -\log\operatorname{softmax}(\tilde{\mathbf{y}})[k] \tag{1}$$

$$\tilde{\mathbf{y}} = \operatorname{fc}(\operatorname{relu}(\tilde{\mathbf{x}})), \tilde{\mathbf{x}} \sim \mathcal{N}(0, I), k \sim \operatorname{Uniform}\{0, \ldots, K\}$$

where $\operatorname{fc}(\cdot)$ is the final (fully connected) layer of the network, $\operatorname{relu}(\tilde{\mathbf{x}}) = \max(\tilde{\mathbf{x}}, \mathbf{0})$ is the Rectified Linear Unit, and $k$ represents a random label from $K$ classes. This represents the case where the model's prediction is uncorrelated with the labels. The actual noise distribution could skew to the left if the model overfits to the noise, and skew to the right if the model predicts a label different from the noisy one. We find that an identity covariance matrix for $\tilde{\mathbf{x}}$ is able to explain the noise distribution; this could result from well-conditioned objectives defined via deep residual networks (He et al., 2015a) and careful initialization (He et al., 2015b). We qualitatively demonstrate the validity of our characterization on CIFAR-100 and ImageNet datasets in Figure 2.

Therefore, we could define a threshold via the $p$-th percentile of $p_n(l)$; it relates to approximately how much examples in $\mathcal{B}$ we would retain if $Q(Y|X)$ is uniform. In Section 4.4, we show that this method is able to identify different percentages of uniform label noise with high precision.

## 2.3 A PRACTICAL ALGORITHM FOR ROBUST TRAINING

We can utilize this implicit regularization effect to remove examples that might harm generalization, leading to *On-the-fly Data Denoising* (ODD), a simple algorithm robust to mislabeled examples:

1. Train all examples with large learning rates for $E$ epochs.

2. Compute the $p$-th percentile of the distribution in Eq. (1), denoted as $T_p$.

3. Remove examples whose average loss of the past $h$ epochs exceeds $T$ from the dataset.

4. Continue training the remaining examples from epoch $E + 1$.

ODD introduces three hyperparameters: $E$ determines the amount of training that separates clean examples from noisy ones; $p$ determines $T_p$ that specifies the trade-off between less noisy examples and more clean examples; $h$ determines the window of averaged loss statistics to reduce variance from data augmentation. We do not explicitly estimate the portion of noise in the dataset, nor do we assume any specific noise model. In fact, the threshold $T_p$ could be used to accurately predict the portion of uniform noise in the dataset, and works quite well even on other types of label noise; we will demonstrate this in Section 4. Moreover, ODD is compatible with existing practices for learning rate schedules, such as stepwise (He et al., 2015a) or cosine (Loshchilov & Hutter, 2016).

## 3 RELATED WORK

**Implicit Regularization of SGD**    The generalization of neural networks trained with SGD depend heavily on learning rate schedules (Loshchilov & Hutter, 2016). It has been proposed that wide local minima[2] could result in better generalization (Hochreiter & Schmidhuber, 1995; Chaudhari et al., 2016; Keskar et al., 2016). Several factors could contribute to wider local optima and better generalization, such as smaller minibatch sizes (Keskar et al., 2016), reasonable learning rates (Kleinberg et al., 2018), and longer training time (Hoffer et al., 2017). Moreover, solutions that are further away from the initialization may lead to wider local minima and better generalization (Hoffer et al., 2017). In the presence of mislabeled examples, changes in optimization landscape (Arpit et al., 2017) could result in bad local minima (Zhang et al., 2016), although it is argued that larger batch sizes could mitigate this effect (Rolnick et al., 2017).

**Training with Mislabeled Examples**    One paradigm to robust training with noisy labels involves estimating the noise distribution (Liu & Tao, 2014) or confusion matrix (Sukhbaatar et al., 2014). Another line of methods propose to identify and clean the noisy examples through predictions of auxillary networks (Veit et al., 2017) or via binary predictions (Northcutt et al., 2017); the noisy labels are either pruned (Brodley et al., 1996) or replaced with model predictions (Reed et al., 2014). Our method is comparable to these approaches, but the key difference is that we leverage the implicit regularization effect of SGD to identify noisy examples. Other approaches propose to reweigh the examples via a pretrained network (Jiang et al., 2017), meta learning (Ren et al., 2018), or surrogate loss functions (Ghosh et al., 2017; Zhang & Sabuncu, 2018). Some methods require a set of trusted examples (Xiao et al., 2015; Hendrycks et al., 2018).

ODD has several appealing properties compared to existing methods. First, the thresholds for classifying mislabeled examples from ODD do not rely on estimations of the noise confusion matrix. Next, ODD does not require additional trusted examples. Finally, ODD removes potentially noisy examples on-the-fly; it has little computational overhead compared to standard SGD training.

## 4 EXPERIMENTS

We evaluate our method on clean and noisy versions of CIFAR-10, CIFAR-100, ImageNet (Russakovsky et al., 2015) and WebVision (Li et al., 2017a) datasets. We use stochastic gradient descent with momentum for training while following standard image preprocessing and data augmentation practices. We do not consider dropout (Srivastava et al., 2014) or model ensembles (Huang et al., 2017) in our experiments. We use $h = 2$ for all our ODD experiments; we observe that having $h \in [2, 5]$ yields similar results.

### 4.1 CIFAR-10 AND CIFAR-100

We first evaluate our method on the CIFAR-10 and CIFAR-100 datasets, which contain 50,000 training images and 10,000 validation images of size $32 \times 32$ with 10 and 100 labels respectively. During training, we follow the data augmentations in (Zagoruyko & Komodakis, 2016), which performs horizontal flips, takes random crops from $40 \times 40$ images padded by 4 pixels on each side, and fills missing pixels with reflections of the original images.

In our experiments, we train the wide residual network architecture (WRN-28-10) in (Zagoruyko & Komodakis, 2016) for 200 epochs with a minibatch size of 128, momentum $0.9$ and weight decay

---

[2]Although the notion of wideness is controversial (Dinh et al., 2017).

$5 \times 10^{-4}$. We consider a *cosine* annealing schedule as described in (Loshchilov & Hutter, 2016) with $T_0 = 200, T_{mult} = 1, \eta_{max} = 0.1, \eta_{min} = 10^{-5}$ (no warm restarts), as we observe this schedule outperforms the traditional stepwise schedules on the clean dataset. We include results for two types of stepwise schedules in Appendix A.1.

### 4.1.1 GENERALIZATION ON INPUT-AGNOSTIC LABEL NOISE

We first consider label noise that are agnostic to inputs. Following Zhang et al. (2016), We randomly replace a $0\%/20\%/40\%$ of the training labels to uniformly random ones, and evaluate generalization error on the clean validation set. We compare with the following baselines: ORACLE, where the model knows the true identity of clean examples and only trains on them; Empirical Risk Minimization (ERM, Goyal et al. (2017)) which assumes all examples are clean; MENTORNET (Jiang et al., 2017), which pretrains an auxiliary model that predicts weights for each example based on its input features; REN (Ren et al., 2018), which optimizes the weight of examples via meta-learning; *mixup* (Zhang et al., 2017), a data augmentation approach that trains neural networks on convex combinations of pairs of examples and their labels; and Generalized Cross Entropy (GCE, Zhang & Sabuncu (2018)) that includes cross-entropy loss and mean absolute error (Ghosh et al., 2017).

Table 1: Minimum validation error throughout training (in percentage).

| Method | CIFAR-10 % of Mislabeled Examples | | | CIFAR-100 % of Mislabeled Examples | | |
|---|---|---|---|---|---|---|
| | 0 | 20 | 40 | 0 | 20 | 40 |
| ORACLE | - | $4.25 \pm 0.05$ | $5.06 \pm 0.05$ | - | $20.42 \pm 0.20$ | $23.64 \pm 0.27$ |
| ERM | $3.73 \pm 0.13$ | $11.53 \pm 0.06$ | $15.57 \pm 0.45$ | $18.41 \pm 0.17$ | $30.38 \pm 0.14$ | $44.35 \pm 0.53$ |
| ODD | $3.80 \pm 0.10$ | $\mathbf{5.33} \pm 0.04$ | $\mathbf{7.16} \pm 0.20$ | $\mathbf{18.19} \pm 0.10$ | $\mathbf{22.79} \pm 0.10$ | $\mathbf{27.61} \pm 0.38$ |
| *mixup* | $\mathbf{3.04} \pm 0.05$ | $6.09 \pm 0.27$ | $9.26 \pm 0.14$ | $18.61 \pm 0.33$ | $28.78 \pm 0.27$ | $40.58 \pm 0.35$ |
| GCE | - | $10.13 \pm 0.20$ | $12.87 \pm 0.22$ | - | $33.19 \pm 0.42$ | $38.23 \pm 0.24$ |
| REN* | - | - | $13.08 \pm 0.19$ | - | - | $38.66 \pm 2.06$ |
| MENTORNET | - | 9 | 23 | - | 28 | 44 |
| MENTORNET* | - | 8 | 11 | - | 27 | 32 |

We report the top-1 validation error in Table 1, where $\star$ denotes methods trained with knowledge of 1000 additional clean labels. Notably, ODD significantly outperforms all other algorithms (except for the oracle) when there is artificial noise, and is on-par with ERM even when there is no artificial noise. On the one hand, this suggests that ODD is able to distinguish the mislabeled examples and improve generalization; on the other hand, it would seem that removing certain examples even in the "clean" dataset does not seem to hinder generalization.

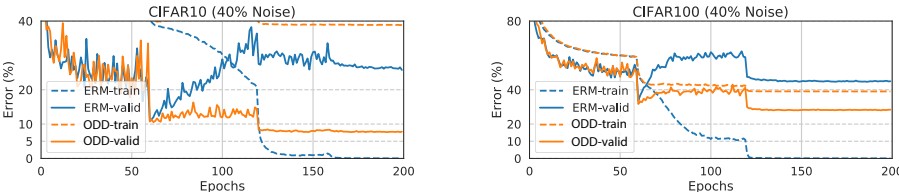

Figure 3: Training curves of ERM and ODD with 40% artificially corrupted labels.

**ODD prevents overfitting to noise**   We compare the learning curves of ERM and ODD in Figure 3 with a stepwise schedule under 40% label corruption. ERM easily overfits the random labels when learning rate decreases, whereas ODD manages to continue improving generalization.

**Mislabeled examples in CIFAR-100**   We run our methods on three random seeds, and find the examples that are considered mislabeled by all the three instances (598 in total); we demonstrate some examples[3] in Figure 1, which contains ambiguous / wrong labels.

---

[3]We include more examples in Appendix A.4.

Table 2: Results on input-dependent label noise experiments.

| % of Corrupted Labels | Method | CIFAR10 | CIFAR100 | Network |
|:---:|:---:|:---:|:---:|:---:|
| 10 | ERM | $4.31 \pm 0.26$ | $19.9 \pm 0.12$ | |
| | ODD | $\mathbf{4.2} \pm 0.02$ | $\mathbf{18.79} \pm 0.05$ | WRN-28-10 |
| 20 | ERM | $4.85 \pm 0.23$ | $24.69 \pm 0.25$ | |
| | ODD | $\mathbf{4.65} \pm 0.19$ | $\mathbf{18.05} \pm 0.01$ | |

### 4.1.2 GENERALIZATION ON INPUT-DEPENDENT LABEL NOISE

Images from some classes could be harder to label correctly than that from other classes. To simulate this, we perform experiments on settings where the label noise only comes from certain types of input data. Specifically, we remove a portion of classes from the dataset (e.g. class 9 in CIFAR-10), and assign the labels of all its examples to the remaining classes randomly (e.g. a class 9 example has a class $0-8$ random label). This reduces the total number of classes, so on the validation set we only consider the classes that are not removed (e.g. classes $0-8$). We compare ERM and ODD on datasets with $10\%$ or $20\%$ of the examples mislabeled, and summarize the results in Table 2. ODD is still able to significantly outperform ERM under such input-dependent noise.

### 4.1.3 GENERALIZATION ON NON-HOMOGENEOUS LABELS

We evaluate ERM and ODD on a setting without mislabeled examples, but the ratio of classes could vary. To prevent the model from utilizing the number of examples in a class, we combine multiple classes of CIFAR-100 into a single class, creating the CIFAR-20 and CIFAR-50 tasks. In CIFAR-50, we combine an even class with an odd class while we remove $c\%$ of the examples in the odd class. In CIFAR-20, we combine 5 classes in CIFAR-100 that belong to the same super-class[4] while we remove $c\%$ of the examples in 4 out of 5 classes. This is performed for both training and validation datasets. Results for ERM and ODD with $p = 10$ and $E = 75$ are shown in Table 3, where ODD is able to outperform ERM in these settings where the input examples are not uniformly distributed.

Table 3: Results on CIFAR-50 and CIFAR-20 tasks.

| Task | Method | $c\%$ of Removed Labels | | | Network |
|:---:|:---:|:---:|:---:|:---:|:---:|
| | | 30 | 50 | 70 | |
| CIFAR-50 | ERM | $21.52 \pm 0.13$ | $22.03 \pm 0.05$ | $22.45 \pm 0.10$ | |
| | ODD | $\mathbf{21.05} \pm 0.16$ | $\mathbf{21.39} \pm 0.18$ | $\mathbf{21.87} \pm 0.08$ | WRN-28-10 |
| CIFAR-20 | ERM | $13.61 \pm 0.20$ | $14.86 \pm 0.08$ | $15.51 \pm 0.32$ | |
| | ODD | $\mathbf{13.37} \pm 0.27$ | $\mathbf{14.63} \pm 0.26$ | $\mathbf{15.44} \pm 0.24$ | |

## 4.2 IMAGENET

We conduct additional experiments on the ImageNet-2012 classification dataset (Russakovsky et al., 2015). The dataset contains 1.28 million training images and 50,000 validation images from 1,000 classes. Input-agnostic random noise of $0\%, 20\%, 40\%$ are considered. We follow standard data augmentation practices during training, including scale and aspect ratio distortions, random crops, and horizontal flips. We only use the center $224 \times 224$ crop for validation.

We train ResNet-50 and ResNet-152 models (He et al., 2015a) with the *cosine* schedule with initial learning rate $0.1$, momentum $0.9$, weight decay $10^{-4}$, 90 training epochs, and report top-1 and top-5 validation errors in Table 4. ODD significantly outperforms ERM in terms of both top-1 and top-5 errors on the 20% and 40% mislabeled examples, while being competitive with the clean dataset.

---

[4]Super-class information is available at `https://www.cs.toronto.edu/~kriz/cifar.html`.

Table 4: Results on ImageNet validation set. MENTORNET results from (Jiang et al., 2017).

| Method | % of Mislabeled Examples | | | | | | Network |
| | 0 | | 20 | | 40 | | |
| | Top-1 | Top-5 | Top-1 | Top-5 | Top-1 | Top-5 | |
|---|---|---|---|---|---|---|---|
| ERM | 23.39 | **6.77** | 26.23 | 8.51 | 29.61 | 10.52 | ResNet-50 |
| ODD ($p = 10$) | **23.37** | 6.94 | **25.05** | **7.89** | **27.51** | **9.25** | |
| ERM | **21.31** | **5.69** | 23.9 | 7.12 | 27.39 | 9.16 | ResNet-152 |
| ODD ($p = 10$) | 21.35 | 5.98 | **22.49** | **6.45** | **25.22** | **7.91** | |
| ERM | 23 | - | - | - | - | - | Inception ResNet-v2 |
| MENTORNET | - | - | - | - | 34.9 | 14.1 | |

Table 5: Validation errors on WebVision / ImageNet validation sets while training on WebVision.

| Method | WebVision | | ImageNet | | Network |
| | Top-1 | Top-5 | Top-1 | Top-5 | |
|---|---|---|---|---|---|
| ERM | $27.90 \pm 0.11$ | $10.94 \pm 0.08$ | $34.71 \pm 0.12$ | $14.87 \pm 0.05$ | ResNet-50 |
| ODD | $\mathbf{27.49} \pm 0.22$ | $\mathbf{10.79} \pm 0.09$ | $\mathbf{34.48} \pm 0.01$ | $\mathbf{14.75} \pm 0.01$ | |
| ERM | 25.71 | **9.30** | 33.62 | 13.97 | Inception ResNet-v2 |
| ODD | **25.38** | 9.37 | **33.27** | **13.68** | |
| MENTORNET | 29.2 | 12.0 | 37.5 | 17.0 | |

## 4.3 WEBVISION – A REAL-WORLD NOISY DATASET

We further verify the effectiveness of our method on a real-world noisy dataset. The WebVision-2017 dataset (Li et al., 2017a) contains 2.4 million of real-world noisy labels, that are crawled from Google and Flickr using the 1,000 labels from the ImageNet-2012 dataset. We train two architectures, ResNet-50 and Inception ResNet-v2 (Szegedy et al., 2016) with the same procedure in the ImageNet experiments, except for Inception ResNet-v2 we train for 50 epochs and use input images of size $299 \times 299$. We use both WebVision and ImageNet validation sets for 1-crop validation, following the settings in Jiang et al. (2017). We do not use a pretrained model or additional labeled data from ImageNet during training.

Our ODD method with $p = 30$ removes in the training set around $9.0\%$ of the total examples with ResNet-50 and $9.3\%$ of the total examples with Inception ResNet-v2 (Szegedy et al., 2016). Table 5 suggests that our method is able to outperform both ERM and MENTORNET when the training dataset is noisy, even as we remove a notable portion of examples. We include more results in Appendix A.3. In comparison, we removed around $1.1\%$ of examples in ImageNet (Table 10, Appendix A.2); this may suggest that WebVision labels are indeed much noisier.

## 4.4 ABLATION STUDIES

**Sensitivity to $p$** We first evaluate noisy ImageNet classification with ResNet-50 where $p \in \{1, 10, 30, 50, 80\}$ and $E = 60$ in Table 6. A higher $p$ includes more clean examples at the cost of involving more noisy examples. In the 20% and 40% noisy cases, the optimal trade-off for generalization is at $p = 10$, yet even when $p = 50$, the validation errors are still significantly better than ERM. When there is no artificial noise, generalization of ODD starts to match that of ERM as $p \geq 10$. Therefore, ODD is not very sensitive to $p$ in these cases, and empirically $p = 10$ represents the best trade-off. We include results for ResNet-152 in Appendix A.2.

**Sensitivity to $E$** We evaluate the validation error of ODD on CIFAR with 20% and 40% input-agnoistic label noise where $E \in \{25, 50, 75, 100, 150, 200\}$ ($E = 200$ is equivalent to ERM). The results in Figure 4 suggest that our method is able to separate noisy and clean examples if $E$ is

Table 6: Ablation studies over the hyperparameter $p$ on ImageNet.

| Method | \% of Mislabeled Examples | | | | | | Network |
| | 0 | | 20 | | 40 | | |
| | Top-1 | Top-5 | Top-1 | Top-5 | Top-1 | Top-5 | |
|---|---|---|---|---|---|---|---|
| ODD ($p=1$) | 23.74 | 7.17 | 25.33 | 8.13 | 27.96 | 9.89 | |
| ODD ($p=10$) | 23.37 | 6.94 | **25.05** | 7.89 | **27.51** | **9.25** | |
| ODD ($p=30$) | 23.48 | 6.89 | **25.05** | **7.77** | 27.75 | 9.4 | ResNet-50 |
| ODD ($p=50$) | 23.52 | 6.89 | 25.31 | 7.92 | 28.63 | 9.4 | |
| ODD ($p=80$) | **23.31** | **6.74** | 25.93 | 8.33 | 29.41 | 10.43 | |

relatively small where the learning rate is high, but is unable to perform well when the learning rate decreases at later stages of the training.

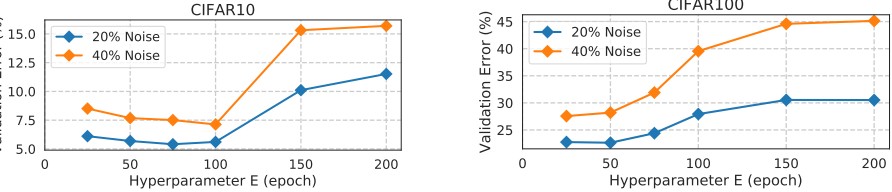

Figure 4: Validation errors of ODD on CIFAR with different values of $E$.

**Sensitivity to the amount of noise** Finally, we evaluate the training error of ODD on CIFAR under input-agnostic label noise of $\{1\%, 5\%, 10\%, 20\%, 30\%, 40\%\}$ with $p = 5$, $E = 50$ or $75$. This reflects how much examples exceed the threshold and are identified as noise at epoch $E$. From Figure 5, we observe that the training error is almost exactly the amount of noise in the dataset, which demonstrates that the loss distribution of noise can be characterized by our threshold regardless of the percentage of noise in the dataset.

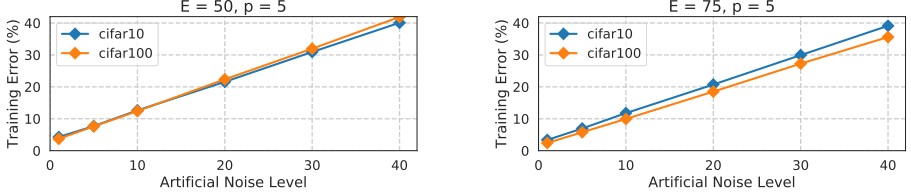

Figure 5: Training errors of ODD on CIFAR with different values of $E$.

## 5 DISCUSSION

We have proposed ODD, a straightforward method for robust training with mislabeled examples. ODD utilizes the implicit regularization effect of stochastic gradient descent to prune examples that potentially harm generalization. Empirical results demonstrate that ODD is able to significantly outperform related methods on a wide range of datasets with artificial and real-world mislabeled examples, maintain competitiveness with ERM on clean datasets, as well as detecting mislabeled examples automatically in CIFAR-100.

The implicit regularization of stochastic gradient descent opens up other research directions for implementing robust algorithms. For example, we could consider using a smaller network to remove examples, removing examples not only once but multiple times, retraining from scratch with the denoised dataset, or other data-augmentation approaches such as *mixup* (Zhang et al., 2017). Moreover, it would be interesting to understand the implicit regularization over mislabeled examples from a theoretical viewpoint.

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

# A  ADDITIONAL EXPERIMENTAL RESULTS

## A.1  CIFAR INPUT-AGNOSTIC NOISE

In addition to the existing experiments, we include results for ORACLE, ERM, ODD with two stepwise annealing schedules. In the stepwise schedules, the learning rate starts from 0.1 and is then divided by 5 after 60, 120, 160 epochs (*stepwise-i*, which is used in Zagoruyko & Komodakis (2016)) or after 100, 150, 175 epochs (*stepwise-ii*). We consider *cosine* schedule in Section 4 because it achieves better generalization performance on the clean dataset.

For the stepwise schedules, we set $E$ to be the epoch at which learning rate begins to decay. For the cosine schedule, we set $E = 100$ for CIFAR-10 and $E = 50$ for CIFAR-100. We set $p = 20$ and $p = 10$ for CIFAR-10 and CIFAR-100 respectively for the clean datasets; $p = 10$ and $p = 5$ for noisy datasets. This is motivated by the fact that CIFAR-10 has less labels, so the threshold has to take into account random labels that happens to be correct.

Table 7: Minimum validation error throughout training (in percentage) on CIFAR10.

| Method | % of Mislabeled Examples | | | Network |
|---|---|---|---|---|
|  | 0 | 20 | 40 |  |
| ORACLE (*cosine*) | - | $4.25 \pm 0.05$ | $5.06 \pm 0.05$ |  |
| ORACLE (*stepwise-i*) | - | $4.45 \pm 0.06$ | $5.29 \pm 0.01$ |  |
| ORACLE (*stepwise-ii*) | - | $4.44 \pm 0.03$ | $5.14 \pm 0.15$ |  |
| ERM (*cosine*) | $3.73 \pm 0.13$ | $11.53 \pm 0.06$ | $15.57 \pm 0.45$ |  |
| ERM (*stepwise-i*) | $3.75 \pm 0.02$ | $8.39 \pm 0.24$ | $10.77 \pm 0.26$ | WRN-28-10 |
| ERM (*stepwise-ii*) | $3.93 \pm 0.10$ | $7.79 \pm 0.03$ | $10.12 \pm 0.27$ |  |
| ODD (*cosine*) | $3.80 \pm 0.10$ | $\mathbf{5.33} \pm 0.04$ | $\mathbf{7.16} \pm 0.20$ |  |
| ODD (*stepwise-i*) | $3.92 \pm 0.09$ | $5.47 \pm 0.16$ | $7.50 \pm 0.15$ |  |
| ODD (*stepwise-ii*) | $4.06 \pm 0.15$ | $5.52 \pm 0.08$ | $7.42 \pm 0.05$ |  |
| MIXUP ($\alpha = 1.0$) | $\mathbf{3.04} \pm 0.05$ | $10.55 \pm 0.31$ | $25.14 \pm 0.08$ |  |
| MIXUP ($\alpha = 8.0$) | $3.39 \pm 0.12$ | $6.09 \pm 0.27$ | $9.26 \pm 0.14$ |  |

Tables 7 and 8 contain summary of the results. The *cosine* learning rate schedule generally out-performs the *stepwise* schedules. We note that for the *stepwise* schedules in ERM, the optimal validation error is achieved when the learning rate just starts to decay to 0.2, after that the model starts to overfit to noise, as demonstrated in Figure 3.

Table 8: Minimum validation error throughout training (in percentage) on CIFAR100.

| Method | % of Mislabeled Examples | | | Network |
|---|---|---|---|---|
|  | 0 | 20 | 40 |  |
| ORACLE (*cosine*) | - | $20.42 \pm 0.20$ | $23.64 \pm 0.27$ |  |
| ORACLE (*stepwise-i*) | - | $20.21 \pm 0.22$ | $24.18 \pm 0.04$ |  |
| ORACLE (*stepwise-ii*) | - | $20.99 \pm 0.25$ | $23.79 \pm 0.05$ |  |
| ERM (*cosine*) | $18.41 \pm 0.17$ | $30.38 \pm 0.14$ | $44.35 \pm 0.53$ |  |
| ERM (*stepwise-i*) | $18.77 \pm 0.33$ | $28.81 \pm 0.48$ | $33.93 \pm 0.26$ | WRN-28-10 |
| ERM (*stepwise-ii*) | $18.80 \pm 0.15$ | $29.28 \pm 0.27$ | $36.72 \pm 0.49$ |  |
| ODD (*cosine*) | $\mathbf{18.19} \pm 0.10$ | $\mathbf{22.79} \pm 0.10$ | $\mathbf{27.61} \pm 0.38$ |  |
| ODD (*stepwise-i*) | $18.82 \pm 0.14$ | $22.95 \pm 0.19$ | $28.01 \pm 0.15$ |  |
| ODD (*stepwise-ii*) | $18.53 \pm 0.10$ | $23.27 \pm 0.34$ | $28.79 \pm 0.03$ |  |
| MIXUP ($\alpha = 1.0$) | $18.61 \pm 0.33$ | $28.78 \pm 0.27$ | $44.43 \pm 0.31$ |  |
| MIXUP ($\alpha = 8.0$) | $20.50 \pm 0.27$ | $28.86 \pm 0.17$ | $40.58 \pm 0.35$ |  |

We evaluate precision and recall for examples classified as noise on CIFAR10 and CIFAR100 for different noise levels (1, 5, 10, 20, 30, 40) in Figure 6. The recall values are around 0.84 to 0.88 where as the precision values range from 0.88 to 0.92. This demonstrates that ODD is able to achieve good precision/recall with default hyperparameters even at different noise levels.

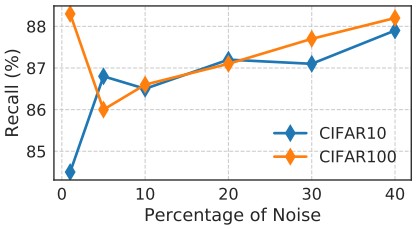 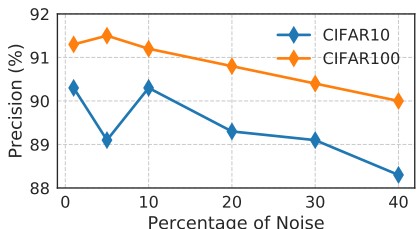

Figure 6: Recall and precision for ODD on CIFAR10 and CIFAR100 with different levels of uniform random noise.

## A.2 IMAGENET ABLATION RESULTS ON RESNET-152

We include the ImageNet ablation experiments on the hyperparameter $p$ on the ResNet-152 architecture in Table 9. Compared to the ResNet-50 experiments, we can draw similar conclusions here: $p = 10$ generally represents the best trade-off. We show the percentage of examples discarded by NOISE CLASSIFIER in Table 10; the percentage of discarded examples by $p = 10$ is very close to the actual noise level. Moreover, the percentage of discarded examples does not vary significantly when we change our architecture from ResNet-50 to ResNet-152.

Table 9: Ablation studies over the hyperparameter $p$ on ImageNet-2012.

| Method | \multicolumn{6}{c}{% of Mislabeled Examples} | Network |
| | 0 | | 20 | | 40 | | |
| | Top-1 | Top-5 | Top-1 | Top-5 | Top-1 | Top-5 | |
|---|---|---|---|---|---|---|---|
| ERM | 21.31 | **5.69** | 23.9 | 7.12 | 27.39 | 9.16 | |
| ODD ($p = 1$) | 21.50 | 6.08 | 22.87 | 6.61 | 25.6 | 8.42 | |
| ODD ($p = 5$) | 21.40 | 5.98 | 22.67 | 6.54 | 25.43 | 8.01 | ResNet-152 |
| ODD ($p = 10$) | 21.35 | 5.98 | **22.49** | **6.45** | **25.22** | **7.91** | |
| ODD ($p = 30$) | 21.45 | 5.78 | 22.68 | 6.52 | 25.77 | 8.24 | |
| ODD ($p = 50$) | **21.04** | 5.7 | 22.78 | 6.5 | 26.73 | 8.9 | |

Table 10: Percentage of example discraded by ODD on ImageNet-2012.

| % Mislabeled | \multicolumn{5}{c}{Hyperparameter $p$} | Network |
| | 1 | 10 | 30 | 50 | 80 | |
|---|---|---|---|---|---|---|
| 0% | 6.2 | 2.6 | 1.1 | 0.7 | 0.5 | |
| 20% | 24.4 | 21.1 | 19.3 | 17.6 | 11.5 | ResNet-50 |
| 40% | 44.8 | 40.3 | 36.2 | 28.1 | 7.8 | |
| 0% | 5.5 | 2.3 | 1.1 | 0.7 | 0.4 | |
| 20% | 23.8 | 20.8 | 19.2 | 17.5 | 0.7 | ResNet-152 |
| 40% | 44.1 | 40.2 | 36.2 | 27.6 | 0.6 | |

## A.3 WebVision Ablation Results

We include the WebVision ablation experiments on the hyperparameter $p$ on the ResNet-50 and Inception ResNet-v2 architecture with $E = 60$ in Tables 11 and 12, where we report top-1 and top-5 validation errors on WebVision and ImageNet validation sets respectively, as well as how many examples are discarded by our method at epoch 60. Similar to the results in ImageNet, generalization performance is generally insensitive to the hyperparameter $p$, except for $p = 1$, which discarded 25.3% of the examples. We use 2 seeds for each experiment setting. Notice that at each $p$, WebVision has more examples discarded compared to ImageNet (with 0% artificial noise), which further suggests that it has more mislabeled examples than ImageNet. Again, the percentage of discarded examples does not vary significantly across different architectures.

Table 11: Ablation studies on WebVision with ResNet-50.

| Method | WebVision | | ImageNet | | % Discarded |
|---|---|---|---|---|---|
| | Top-1 | Top-5 | Top-1 | Top-5 | |
| ERM | $27.90 \pm 0.11$ | $10.94 \pm 0.08$ | $34.71 \pm 0.12$ | $14.87 \pm 0.05$ | - |
| ODD ($p = 1$) | $28.03 \pm 0.03$ | $11.45 \pm 0.05$ | $34.68 \pm 0.03$ | $15.05 \pm 0.08$ | 25.3 |
| ODD ($p = 10$) | $\textbf{27.45} \pm 0.01$ | $11.07 \pm 0.01$ | $34.47 \pm 0.01$ | $14.80 \pm 0.01$ | 15.1 |
| ODD ($p = 30$) | $27.49 \pm 0.22$ | $10.79 \pm 0.09$ | $34.48 \pm 0.01$ | $\textbf{14.75} \pm 0.01$ | 9.0 |
| ODD ($p = 50$) | $27.59 \pm 0.01$ | $\textbf{10.77} \pm 0.01$ | $34.40 \pm 0.03$ | $14.81 \pm 0.02$ | 5.8 |
| ODD ($p = 80$) | $27.53 \pm 0.02$ | $10.82 \pm 0.01$ | $\textbf{34.27} \pm 0.04$ | $14.81 \pm 0.02$ | 2.5 |

Table 12: Ablation studies on WebVision with Inception ResNet-v2.

| Method | WebVision | | ImageNet | | % Discarded |
|---|---|---|---|---|---|
| | Top-1 | Top-5 | Top-1 | Top-5 | |
| ERM | 25.71 | 9.3 | 33.62 | 13.97 | - |
| ODD ($p = 1$) | 25.99 | 10.07 | 34.23 | 14.6 | 25.7 |
| ODD ($p = 10$) | 25.69 | 9.45 | 33.91 | 14.14 | 15.2 |
| ODD ($p = 30$) | **25.38** | 9.37 | **33.27** | **13.68** | 9.3 |
| ODD ($p = 50$) | 25.57 | **9.22** | 33.42 | 13.79 | 6.1 |
| ODD ($p = 80$) | 25.50 | 9.21 | **33.27** | 13.76 | 2.8 |

### A.4  IMAGES IN CIFAR-100 CLASSIFIED AS NOISE

We display the examples in CIFAR-100 training set for which our ODD methods identify as noise across 3 random seeds. One of the most common label such examples have is "leopard"; in fact, 21 of 50 "leopard" examples in the training set are perceived as hard, and we show some of them in Figure 7. It turns out that a lot of the "leopard" examples contains images that clearly contains tigers and black panthers (CIFAR-100 has a label corresponding to "tiger").

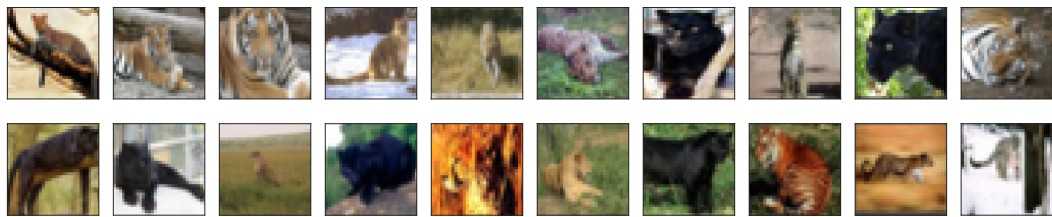

Figure 7: Examples with label "leopard" that are classified as noise.

We also demonstrate random examples from the CIFAR-100 that are identified as noise in Figure 8 and those that are not identified as noise in Figure 9. The examples identified as noise often contains multiple objects, and those not identified as noise often contains only one object that is less ambiguous in terms of identity.

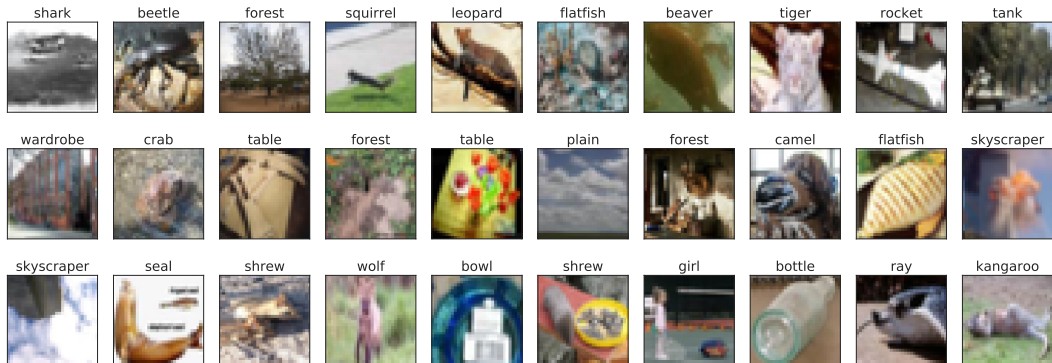

Figure 8: Random CIFAR-100 examples that are classified as noise.

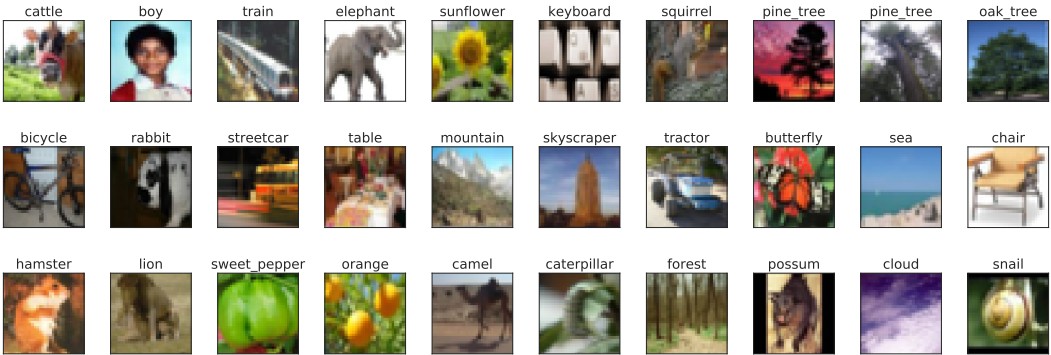

Figure 9: Random CIFAR-100 examples that are not classified as noise.

