# OpenReview forum: "Better Generalization with On-the-fly Dataset Denoising"
_ICLR.cc/2019/Conference_

### Official Review · AnonReviewer1 · 2018-11-02
**BETTER GENERALIZATION WITH ON-THE-FLY DATASET DENOISING**

**Rating:** 6
**Confidence:** 5

**Review:**

This paper presents ODD, a method that rejects incorrectly labeled / noisy examples from training on the fly. The motivation is sound, that with the capacity of modern neural networks, it's easy to memorize the mislabeled data and thus hurt generalization. If we could reject such mislabeled data, we may be able to get a more generalizable model. The authors made an observation that when training with large learning rate, examples with correct labeling and incorrect labeling exhibits different loss distributions. The authors further noticed that the loss distribution of incorrectly labeled examples can be simulated using eq.(1). Therefore, by setting a threshold that corresponds to a percentile of the incorrectly labeled loss distribution, the authors are able to reject incorrect examples.

Some comments:
1. Eq.(1) basically assumes all the noise is uniformly distributed among classes. What if only 2 classes are easily mislabeled while others are fine?
2. Section 4.1.3 and Section 4.4 Sensitivity to Noise are confusing. Please clarify the importance and rationale for such analysis.
3. Cosine schedule is used in the experiments. However, since the method does not work well with small learning rate, why not using a fixed large learning rate and decrease it after noise rejection? Also, in section 4.4 Sensitivity to E, the analysis of the sensitivity to the number of epochs is coupled with a changing learning rate. It would be better to see an experiment with the two decoupled.
4. The loss of an example is averaged over h epochs. It will better to clarify how the simulated distribution generated in such case since the distribution is dependent on fc(.), which is changed between two epochs.
5. Except for the first experiment, all other experiments are only compared with ERM, the vanilla algorithm. It would be better to show a comparison with other methods.
6. Please show a precision/recall of the examples that are marked as "noise" by the method.
7. I assume this method will remove a lot of hard examples. How does this affect training? Does this make the network more error-prone to harder instances?

---

> ### Author Response · Authors · 2018-11-16
> **Addressing individual comments**
>
> Thanks for the review! One of our goals in this paper is to demonstrate that with the right principles, simple methods (ODD) can perform much better than complicated ones (based on Meta Learning). We answer Q5 first to address the comparison with other methods.
>
> Q5: Performance comparison with other methods.
>
> We compared the numbers reported by MentorNet on other experiments. MentorNet is the state-of-the-art method to address robust deep learning (in ICML18). However, our results show that MentorNet can be outperformed by switching to a cosine learning rate schedule alone. Other related methods are already outperformed by MentorNet from the comparisons in the MentorNet paper.
>
> MentorNet requires tuning of a wide range of hyperparameters, including the pretrained denoising schedule, the architecture of the Mentor network, and even the individual dropout ratio for each epoch of training. This is much more complicated compared to ODD, which makes little changes to standard ERM training, and only introduces a few hyperparameters that are interpretable and easy to tune.
>
> Therefore, we believe that ODD can serve as a simple but effective baseline for future work in robust supervised deep learning. Future work in this direction can benefit from the implicit regularization principles found in this paper.
>
> Q1: What if the actual noise model is not uniformly random?
>
> Thanks for mentioning this! In real applications, the actual noise model is unlikely to be uniformly random; in fact, we have no information about the noise model at all! ODD makes no assumptions about the noise transition matrix or the amount of noise.
>
> We assume uniformly random noise because of two reasons. First, this case is very easy to analyze and simulate, and we can evaluate the performance easily across multiple noise levels. Second, this noise has higher entropy than any other type of noise, making it easier to separate than clean labels. We use this analysis to select suitable threshold values that balances precision and recall reasonably as controlled by hyperparameter p.
>
> Q2: Clarify the rationale for such analysis
>
> Section 4.1.3 evaluates ODD’s performance when the dataset inputs are imbalanced, which is very common in real-world datasets (such as WebVision). By combining several labels into one, we prevent any method that tries to balance the dataset simply by counting the number of labels. CIFAR50 considers randomly merging the labels, and CIFAR20 considers merging the labels that are semantically related. Our positive results demonstrate that ODD is capable of dealing with unbalanced datasets as well.
>
> Section 4.4 evaluates ODD’s sensitivity to the hyperparameters (p, E) as well as the portion of uniform noise in the dataset. We demonstrate that ODD is generally insensitive to changes of hyperparameters, and that ODD performs well under all the uniform noise levels. This suggests that we could bypass the process of estimating the amount of noise (which is unknown in realistic cases) in the dataset and tune (p, E) directly for ODD. While uniform noise is unrealistic, the amount of data subject to this type of noise can be controlled, unlike real-world noise.
>
> Q3: Why not use large learning rate and decay? Why not decouple E and learning rate?
>
> In Figure 3 and Table 7, we demonstrate the performance of the stepwise learning rate schedule. For the stepwise learning rate schedule, we used a large learning rate at first, and set E to the epoch where we first decrease the learning rate. From Figure 3, the smaller learning rate will start to overfit immediately if ODD is not performed. This forces E to be related to the learning rate schedule for the stepwise case. Moreover, the results from cosine outperform stepwise consistently, so we use cosine as a strong baseline to compare with.

---

> ### Author Response · Authors · 2018-11-16
> **Addressing individual comments (continued)**
>
> Q4: How to merge statistics from multiple epochs?
>
> We merely take the average over the log-likelihood loss of the same example (which could be different due to data augmentation). The loss statistics does not change significantly over one epoch of training.
>
> Q6: Precision-Recall for examples classified as noise.
>
> We evaluated precision and recall for examples classified as noise on CIFAR10 and CIFAR100 for different noise levels (1, 5, 10, 20, 30, 40), with hyperparameter settings in Appendix A.1. The recall values are around 0.87 where as the precision values range from 0.88 to 0.92. This demonstrates that ODD is able to achieve good precision/recall with default hyperparameters at different noise levels. We update these results in the Appendix.
>
> Q7: What is the amount of hard example removed? How does it affect error on “harder” instances?
>
> In Figure 5, Table 10 and Table 11, we demonstrate the amount of examples classified as “hard” / “noisy”. The amount varies, but generally is smaller on clean datasets than noisy datasets (15% on WebVision, and 2.5% on ImageNet).
>
> The error on there harder instances would definitely be higher, since ODD stopped training them after E epochs. However, we demonstrate that this could even improve generalization, since these “hard” examples could be mislabeled, and therefore not worth learning in the first place. This is in contrast to other approaches such as “hard example mining” which assumes the “hard examples” are not mislabeled.

---

### Official Review · AnonReviewer3 · 2018-11-03
**Interesting paper with sufficient empirical experiments**

**Rating:** 6
**Confidence:** 3

**Review:**

The paper aims to remove potential examples with label noise by discarding the ones with large losses in the training procedure. The idea also applies to the setting where instances may contain large noise. The proposed method may have an implicit trade-off between the robust to label noise and feature noise, which explains why the proposed method also has good performances on instance-dependent label noise. The paper is well-written and has sufficient experiments.

The discussions in Section 2.2 is unclear for me. What is "p_n(\ell)"? What is "p-th percentile of a distribution"? How reasonable thresholds are derived for the uniform label noise? Why the method will generalize to other types of label noise?

===
After reading the rebuttal, it is still unclear of how to determine the thresholds for finding incorrect labels. The authors empirically demonstrated a procedure to statistically find a threshold under the assumption that the label noise is uniform. However, theoretical guarantees are lacking. The extension to other types of label noise is also very intuitive. Although the proposed method is simple and effective, the lack of an effective method for choosing the threshold is a major concern for real-world applications. Are there some other ways to determine the threshold? For example, cross-validation method?

---

> ### Author Response · Authors · 2018-11-16
> **Uniform random noise is used to analyze the regularization effect of SGD, which generalizes across multiple types of noise.**
>
> Thanks for your review!
>
> Q: What is “p_n(\ell)”
> This is “the estimated (negative log-likelihood) loss distribution of uniform label noise”, as described before Equation (1).
>
> Q: "p-th percentile of a distribution D"
> This means “p” percent of the samples “s” from the distribution D are lower than the value “V”, so P(s < V) = p.
>
> Q: How reasonable thresholds are derived for the uniform label noise?
> The reason why we used uniform label noise is because this is the simplest type of noise. Moreover, the entropy of this label noise is highest, so any other type of label noise would have smaller entropy. As a consequence, they are harder to separate than uniform label noise. Therefore, the threshold for other types of noise (including real world ones), should be higher than the 1/K-th percentile of “p_n(\ell)”, where K denotes the number of labels (to reduce false positives), motivating the percentile hyperparameter “p”.
>
> Q: Why does the method generalizes to other types of noise?
> In our analysis, we assume a label is either “correct / clean” or “incorrect / noise”. We wish to demonstrate that by leveraging the implicit regularization effect of large learning rate SGD, it is easier to separate the “correct” and “incorrect” labels.
>
> The uniformly random noise is used to gain “ground-truth” knowledge about the “correct / incorrect” information, since we do not have this information in real-world datasets. For other types of noise, only the distribution of “correct” and “incorrect” labels is changed; this should not affect SGD’s ability to separate them.
>
> The experiments on CIFAR100 "clean" dataset also supports this claim (Figure 1), as the examples that are treated as "incorrect" by ODD do contain more "incorrect" labels.

---

> ### Author Response · Authors · 2018-12-06
> **Cross validation could help determine the optimal trade-off between accuracy and robustness**
>
> As you suggested, cross-validation on a clean validation dataset could be used to determine which hyperparameter works well, and is generally used to determine other hyperparameters in practice (such as neural network architectures and learning rate schedules).
>
> We used the uniform label noise case (with maximum label entropy) to demonstrate the "range" of thresholds that the user can choose; any threshold beyond that is infeasible. Since we do not know the amount of noise or the type of noise in practice, this threshold has to be determined via cross-validation to find the optimal trade-off between accuracy and robustness (both could affect the validation error).

---

### Official Review · AnonReviewer2 · 2018-11-03
**Simple and interesting work**

**Rating:** 5
**Confidence:** 4

**Review:**

Thanks for the rebuttal. But, I am still not very convinced with the proposed results. For CIFAR-100 (0%), you get about 0.2% gain, for ImageNet (0%), you get about 0.2% loss in top-5 accuracy, and for WebVision, you get about 0.3% gain. I am not sure whether you can call these as statistically significant gains. I believe such gain/loss can be obtained with many other tweaks, such as the learning rate scheduling, as the authors have done.

I believe extensive testing the proposed method on many real noisy datasets, not the synthetically generated ones, and showing the consistent gains would much strengthen the paper. But, at the current version, the only such result is Table 5, which is, again, not very convincing to me.

So, I still keep my rating.

=======

Summary:

The authors propose a simple empirical method for cleaning the dataset for training. By using the implicit regularization property of SGD-based optimization method, the authors come up with a method of setting a threshold for the training loss statistics such that the examples that show losses above the threshold are regarded as noisy examples and are discarded. Their empirical results show that ODD (their method) can outperform other baselines when artificial random label noise is injected. They also show ablation studies on the hyperparameters and show the final result seems to be robust to those parameters.

Pros:
- The method is very simple
- The empirical results, particularly on the synthetic noisy training data, seems to be encouraging.
- The ablation study argues that the method is robust to the hyperparameters, p, E, and h.

Cons:
- I think the results remains to be highly empirical. While it is interesting to see the division of the loss statistics in Figure 2, I am not very convinced about the real usage of the proposed method. The result in Table 5 shows that ODD can outperform ERM for real world datasets, but the improvement seems to be marginal. Moreover, the hyperparameter p was set to 30 for that experiment, but how did the authors choose that parameter? Clearly, if you choose wrong p, I think the performance will degrade, and it is not clear how you can choose p in real applications. The ablation studies are only with synthetic noisy label data, so I think the result is somewhat limited.
-

I think the paper shows interesting results, but my concern is that it seems to be quite empirical. The positive results are particularly on the synthetic data case.

---

> ### Author Response · Authors · 2018-11-16
> **Simple baselines like ODD could serve as drop-in replacement for ERM**
>
> Thank you for your review. The goal of our paper is to show that in the case of deep learning with noisy supervision, a simple method, ODD, can significantly outperform sophisticated alternatives, such as MentorNet and Ren’s Meta Validation Optimization (both in ICML18). We believe ODD can serve as a drop-in replacement for ERM for robust deep learning, and the findings could lead to more theoretical insights.
>
> Q: Improvement on real datasets seems marginal.
>
> We respectfully disagree: while the improvement on ODD over ERM on WebVision seems marginal (around .5% top-1 error), the improvement of Inception ResNet-v2 over ResNet-50 is also only 1-2%. However, Inception ResNet-v2 increases training time from 2 days to 6 days (with 8 V100 GPUs). ODD, on the other hand, drops about 15% of the data and thus saves training time for both models.
>
> ODD can also be easily implemented, making it appealing to practitioners who train their models on real-world datasets. These datasets can contain an unknown source of label noise which can hurt generalization.
>
> Despite its simplicity, ODD outperforms all the state-of-the-art methods on datasets with various types of label noise, and is competitive to the state-of-the-art on clean datasets. This includes CIFAR100 (“clean” version) as well as WebVision (a dataset based on noisy web supervision). ODD also demonstrates that even datasets which are assumed to be “clean”, such as CIFAR100, actually contain a lot of mislabeled examples. Our method could alleviate the process of tedious label cleaning, as a lot of the noisy labels are automatically detectable.
>
>
> Q: How to choose p?
> In Table 11 (appendix), we show that the performance of ODD is not much affected for various settings of p = 10, 30, 50, 80 on WebVision; similar arguments could be made for clean ImageNet. Even as ODD requires tuning p, it bypasses the more sophisticated hyperparameters, such as the amount of noise / weight between different objectives / extra MentorNet architecture proposed by previous methods. In practice we find that p ranging from 10 to 50 gives reasonable results.
>
> Q: Ablation studies only with synthetic noisy label data.
>
> Table 11 & 12 show results for various settings of p on WebVision - a real-world dataset with noisy labels. It would be impossible to measure the amount of “real-world noise” let alone control the amount in an ablation study. We therefore conducted experiments on synthetic data in order to precisely control the amount of noise. The same methodology is used in both the MentorNet paper and Ren’s ICML18 paper.
>
> Therefore, we believe that practitioners can apply ODD as a drop-in replacement for ERM for their (potentially noisy) datasets, so it should be of interest to ICLR. The implicit regularization effect can be used to filter out potentially mislabeled data and to improve generalization. Explaining why large learning rates would have this effect is also interesting, but could require more assumptions than is realistic at this point. We believe this is an interesting open problem for the theoretical community.

---

> ### Author Response · Authors · 2018-12-07
> **We evaluated standard deviations for most datasets**
>
> We evaluated all methods with multiple runs and compute the mean and standard deviation of accuracy (except WebVision, we do not have enough resources). The standard deviation is relatively small compared to the empirical gains. Moreover, we use the same training hyperparameters for ODD and ERM in all the comparisons, so the improvement is not due to random artifacts such as learning rate schedule.
>
> We argue that the synthetic label noise would of practical relevance when the user is faced with potential "data poisoning" by adversaries, who would create adversarial labels to hinder the generalization performance. The synthetic noise experiments are meaningful in showing that ODD is much more robust compared to existing methods, while still being comparable to ERM on clean datasets. There are a lot of relevant work on using synthetic label noise:
>
> Generalized Cross Entropy Loss for Training Deep Neural Networks with Noisy Labels, NIPS 2018
> Masking: A New Perspective of Noisy Supervision, NIPS 2018
> Co-teaching: Robust training of deep neural networks with extremely noisy labels, NIPS 2018
>
> Thanks for the suggestion on real world noisy datasets. Is there any such "noisy" dataset that you find would be relevant? We will add these experiments in the final version.

---

### Public Comment · ~Lu_Jiang1 · 2018-12-03
**Impressive Results**

Dear authors,

Thank you for the paper. This paper presents a simple method but achieves very impressive empirical results. It is examined in a series of datasets of synthetic and real-world noise.

Based on your results, I found our hyper-parameters setting on ImageNet and WebVision is far from optimal. Thanks for the insight. On the other hand, I found it is interesting that your base model is more robust to the noisy labels than ours, which makes it difficult for you to further improve. This might be because of the learning rate, dropout strategy, pre-processing, or other implementation details.

I have a suggestion (please correct me if I am wrong):
Using the moving average to compute the p-th percentile is a key technique proposed in (Jiang et al 2017). It would be nice this can be briefly mentioned in Section 2.3.

Thank you for the good work and look forward to reading its final version.

---

> ### Author Response · Authors · 2018-12-03
> **Thank you**
>
> Thank you very much for your comments!
>
> Indeed, some "standard" training techniques in the past few years could be overshadowed by new but less popular ones.
>
> For example, we find that the SGDR cosine schedule significantly outperforms any other stepwise schedule we tried (which was initially proposed for ResNet training), on all the datasets. We believe that using strong baselines with good regularization techniques is important, since they allow us to demonstrate the ability to use ODD to replace standard ERM in practice.
>
> We did not use dropout in the experiments, although this could be helpful. We believe the greatest performance gain comes from using SGDR. MentorNet could also benefit from using SGDR; we will emphasize this in the experiments.
>
> We will mention your technique in the final version.

---

> > ### Public Comment · ~Lu_Jiang1 · 2018-12-06
> > **Thank you**
> >
> > Awesome. Thank you for the response.
> > I hope this paper can get in and we can have some in-depth discussions at the conference.

---

### Meta-Review · Area_Chair1 · 2018-12-12
**Arguable assumption and ad-hoc approach**

**Confidence:** 4
**Recommendation:** Reject

**Metareview:**

The paper aims to clean data samples with label noise in the training procedure.

The reviewers and AC note the following potential weaknesses: (1) the assumption of uniform noise, which is not the case in practice, (2) marginal gains under real-world datasets and (3) highly empirical and ad-hoc approach.

AC thinks the proposed method has potential and is interesting, but decided that the authors need more significant works to publish the work.